# Evaluating Gender Bias in Natural Language Inference

## Abstract

Gender-bias stereotypes have recently raised significant ethical concerns in natural language processing. However, progress in detection and evaluation of gender-bias in natural language understanding through inference is limited and requires further investigation. In this work, we propose an evaluation methodology to measure these biases by constructing a challenge task which involves pairing gender neutral premise against gender-specific hypothesis. We use our challenge task to investigate state-of-the-art NLI models on the presence of gender stereotypes using occupations. Our findings suggest that three models (BERT, RoBERTa, BART) trained on MNLI and SNLI data-sets are significantly prone to gender-induced prediction errors. We also find that debiasing techniques such as augmenting the training dataset to ensure a gender-balanced dataset can help reduce such bias in certain cases.

## 1 Introduction

Machine learning algorithms trained in natural language processing tasks have exhibited various forms of systemic racial and gender biases. These biases have been found to exist in many subtasks of NLP, ranging from learned word embeddings (Bolukbasi et al., 2016; Brunet et al., 2019), natural language inference (He et al., 2019a), hate speech detection (Park et al., 2018), dialog (Henderson et al., 2018; Dinan et al., 2019), and coreference resolution (Zhao et al., 2018b). This has prompted a large area of research attempting to evaluate and mitigate them, either through removal of bias introduction in dataset level (Barbosa & Chen, 2019), or through model architecture (Gonen & Goldberg, 2019), or both (Zhou & Bansal, 2020).

Specifically, we revisit the notion of detecting gender-bias in Natural Language Inference (NLI) systems using targeted inspection. NLI task constitutes of the model to understand the inferential relations between a pair of sentences (premise and hypothesis) to predict a three-way classification on their entailment, contradiction or neutral relationship. NLI requires representational understanding between the given sentences, hence its critical for production-ready models in this task to account for less to no perceivable stereotypical bias. Typically, NLI systems are trained on datasets collected using large-scale crowd-sourcing techniques, which has its own fair share of issues resulting in the introduction of lexical bias in the trained models (He et al., 2019b; Clark et al., 2019). Gender bias, which is loosely defined by stereotyping gender-related professions to gender-sensitive pronouns, have also been found to exist in many NLP tasks and datasets (Rudinger et al., 2017; 2018).

With the advent of large-scale pre-trained language models, we have witnessed a phenomenal rise of interest in adapting the pre-trained models to downstream applications in NLP, leading to superior performance (Devlin et al., 2019; Liu et al., 2019; Lewis et al., 2019). These pre-trained models are typically trained over a massive corpus of text, increasing the probability of introduction of stereotypical bias in the representation space. It is thus crucial to study how these models reflect the bias after fine-tuning on the downstream task, and try to mitigate them without significant loss of performance.

The efficacy of pre-trained models on the downstream task also raises the question in detecting and mitigating bias in NLP systems - *is the data or the model at fault?*. Since we fine-tune these pre-trained models on the downstream corpus, we can no longer conclusively determine the source of the bias. Thus, it is imperative to revisit the question of detecting the bias from the final sentence representations. To that end, we propose a challenge task methodology to detect stereotypical gender bias

in the representations learned by pre-trained language models after fine-tuning on the natural language inference task. Specifically, we construct targeted sentences inspired from Yin et al. (2019), through which we measure gender bias in the representation space in the lens of natural language inference. We evaluate a range of publicly available NLI datasets (SNLI (Bowman et al., 2015), MNLI (Williams et al., 2018), ANLI (Nie et al., 2020) and QNLI (Rajpurkar et al., 2016),) and pair them with pre-trained language models (BERT (Devlin et al. (2019)), RoBERTa (Liu et al. (2019)) and BART (Lewis et al. (2019))) to evalute their sensitivity to gender bias. Using our challenge task, we detect gender bias using the same task the language models are fine-tuned for (NLI). Our challenge task also highlights the direct effect and consequences of deploying these models by testing on the same downstream task, thereby achieving a thorough test of generalization. We posit that a biased NLI model that has learnt gender-based correlations during training will have varied prediction on two different hypothesis differing in gender specific connotations.

Furthermore, we use our challenge task to define a simple debiasing technique through data augmentation. Data augmentation have been shown to be remarkably effective in achieving robust generalization performance in computer vision (DeVries & Taylor, 2017) as well as NLP (Andreas, 2020b). We investigate the extent to which we can mitigate gender bias from the NLI models by augmenting the training set with our probe challenge examples. Concretely, our contributions in this paper are:

- We propose an evaluation methodology by constructing a challenge task to demonstrate that gender bias is exhibited in state-of-the-art finetuned Transformer-based NLI model outputs (Section 3).

- We test augmentation as an existing debiasing technique and understand its efficacy on various state-of-the-art NLI Models (Section 4). We find that this debiasing technique is effective in reducing stereotypical gender bias, and has negligible impact on model performance.

- Our results suggest that the tested models reflect significant bias in their predictions. We also find that augmenting the training-dataset in order to ensure a gender-balanced distribution proves to be effective in reducing bias while also maintaining accuracy on the original dataset.

## 2 PROBLEM STATEMENT

MNLI (Williams et al. (2018)) and SNLI (Bowman et al. (2015)) dataset can be represented as $D < P, H, L >$ with $p \, \epsilon \, P$ as the premise, $h \, \epsilon \, H$ as the hypothesis and $l \, \epsilon \, L$ as the label (entailment, neutral, contradiction). These datasets are created by a crowdsourcing process where crowd-workers are given the task to come up with three sentences that entail, are neutral with, and contradict a given sentence (premise) drawn from an existing corpus.

Can social stereotypes such as gender prejudice be passed on as a result of this process? To evaluate this, we design a challenge dataset $D' < P, F, M >$ with $p \, \epsilon \, P$ as the premise and $f \, \epsilon \, F$ and $m \, \epsilon \, M$ as two different hypotheses differing only in the gender they represent. We define gender-bias as the representation of $p$ learned by the model that results in a change in the label when paired with $f$ and $m$ separately. A model trained to associate words with genders is prone to incorrect predictions when tested on a distribution where such associations no longer exist.

In the next two sections, we discuss our evaluation and analysis in detail and also investigate the possibilities of mitigating such biases.

## 3 MEASURING GENDER BIAS

We create our evaluation sets using sentences from publicly available NLI datasets. We then test them on 3 models trained on MNLI and SNLI datasets. We show that a change in gender represented by the hypothesis results in a difference in prediction, hence indicating bias.

## 3.1 DATASET

We evaluate the models on two evaluation sets: in-distribution $I$, where the premises $p$ are picked from the dataset (MNLI (Williams et al. (2018)), SNLI (Bowman et al. (2015))) used to train the models and out-of-distribution $O$, where we draw premises $p'$ from NLI datasets which are unseen to the trained model. For our experiments in this work we use ANLI (Nie et al. (2020)) and QNLI (Rajpurkar et al. (2016)) for out-of-distribution evaluation dataset creation. Each premise, in both $I$ and $O$, is evaluated against two hypothesis $f$ and $m$, generated using templates, each a gender counterpart of each other. Statistics of the datasets are shown in Table 1.

| Dataset | # instances | Source of premise |
|---|---|---|
| In-distribution Evaluation Set (MNLI) | 1900 | MNLI original dataset |
| In-distribution Evaluation Set (SNLI) | 1900 | SNLI original dataset |
| Out-of-distribution Evaluation Set (ANLI + QNLI) | 3800 | (ANLI + QNLI) original dataset |

Table 1: Statistics of evaluation sets designed for evaluating gender bias

**Premise:** To measure the bias, we select 38 different occupations to include a variety of gender distribution characteristics and occupation types, in correspondence with US Current Population Survey [1] (CPS) 2019 data and prior literature (Zhao et al. (2018a)). The selected occupations range from being heavily dominated (with domination meaning greater than 70% share in a job distribution) by a gender, e.g. nurse, to those which have an approximately equal divide, e.g. designer. The list of occupations considered can be found in Appendix (A.3).

From our source of premise, as mentioned in Table 1, we filter out examples mentioning these occupations. Next, we remove examples that contain gender specific words like *"man", "woman", "male", "female"* etc. On our analysis, we found out that models were sensitive to names and that added to the bias. Since, in this work, our focus is solely on the bias induced by profession, we filtered out only the sentences that didn't include a name when checked through NLTK-NER (Bird et al. (2009)).

We equalize the instances of the occupations so that the results are not because of the models' performance on only a few dominant occupations. For this, we used examples from occupations with larger share in the dataset and used them as place-holders to generate more sentences for occupations with lesser contribution. Examples of this can be seen in Table 2.

| Source Occupation | Original Premise | Target Occupation | Modified Premise |
|---|---|---|---|
| Nurse | A nurse is sitting on a bench in the park. | Teacher | A teacher is sitting on a bench in the park. |
| Janitors | Janitors are doing their job well. | Teacher | Teachers are doing their job well. |
| Carpenter | A carpenter is walking towards the store. | Baker | A baker is walking towards the store. |

Table 2: The source sentence acts as a placeholder and we replace the source occupation with the target occupation to generate a new sentence. This is done to augment our evaluation set to ensure equal number of premises for all 38 occupations.

Following this methodology we generated datasets consisting of equal number of sentences from each of the occupations to act as the premise of our evaluation sets. The sentences were intended to be short (max. 10 words) and grammatically simple to avoid the inclusion of other complex words that may affect the model's prediction. We verified that each sentence included is gender neutral and does not seek to incline to either male or female in itself.

**Hypothesis:** We use templates T to generate gender-specific hypothesis, as shown in Table 3. Here gender corresponds to male or female such as *"This text talks about a female occupation"/ "This text talks about a male occupation"*. We focus on making the template sentences help discern the gender bias in the NLI models. We also vary the structure of these templates to ensure that the results are purely based on the bias and are not affected by the syntactic structure of the hypothesis. We consider a hypothesis "pro-stereotypical" if it aligns with society's stereotype for an occupation, e.g. "female nurse" and anti-stereotypical if otherwise.

Admittedly, more natural hypotheses can be created through crowd-sourcing, but in this work we generate only the baseline examples and we leave more explorations in this regard as a future work.

---

[1] Labor Force Statistics from the Current Population Survey( https://www.bls.gov/cps/cpsaat11.htm)

| Hypothesis Templates |
|---|
| This text speaks of a [gender] profession |
| This text talks about a [gender] occupation |
| This text mentions a [gender] profession |

Table 3: Templates used for generation of hypothesis. Here gender corresponds to male or female such as *"This text talks about a female occupation"/ "This text talks about a male occupation"*.

## 3.2 EXPERIMENTS

The key idea of our experiments is to test our null hypothesis according to which the difference between predicted entailment probabilities, $P_f$ and $P_m$, on pairing a given premise $p$ with female hypothesis $f$ and male hypothesis $m$ respectively, should be 0.

We test our evaluation sets for each of the models mentioned in Section 3.2.1. For every sentence used as a premise, the model is first fed the female specific hypothesis, $f$, followed by its male alternative, $m$.

A typical RTE task would predict one of the three labels - entailment, neutral and contradiction - for the given pair of premise and hypothesis. For this experiment we investigate if the model predicts the textual entailment to be "definitely true" or "definitely false". Hence, we convert our problem into a binary case: "entailment" vs. "contradiction" thus scraping the logit for the "neutral" label and taking a softmax over the other two labels.

### 3.2.1 MODELS AND TRAINING DETAILS

Transformer models pretrained on large dataset have shown state-of-the art performance in the task of RTE for various NLI datasets. In this work, we use three models that have been widely used both in research and production:

- **BERT** (Devlin et al. (2019)) is the Bidirectional Encoder Representations from Transformers, pretrained using a combination of masked language modeling objective and next sentence prediction on a large corpus comprising the Toronto Book Corpus and Wikipedia. It obtained state-of-art results for recognition of textual entailment on both MNLI and SNLI datasets when it was first released.

- **RoBERTa** follows a Robustly Optimized BERT Pretraining Approach (Liu et al. (2019)). It improves on BERT by modifying key hyperparameters in BERT, and training with much larger mini-batches and learning rates.

- **BART** (Lewis et al. (2019)), a denoising autoencoder for pretraining sequence-to-sequence models, is trained by (1) corrupting text with an arbitrary noising function, and (2) learning a model to reconstruct the original text. It matches RoBERTa's performance on natural language understanding tasks.

| Model | Configuration |
|---|---|
| BERT | bert-base-uncased |
| RoBERTa | roberta-base-v2 |
| BART | facebook/bart-base |

Table 4: Configurations of the models tested for gender-bias

**Hyperparameters:** We fine-tune above models on MNLI and SNLI datasets each generating a total of 6 models (3 for each dataset) to test our evaluation sets on. The pretrained configuration used for each of the models is mentioned in Table 4. We train all models using AdamW optimizer with $\beta1$ = 0.9, $\beta2$ = 0.999 and L2 weight decay of 0.01. A learning rate of 1e-5 was used for RoBERTa and 2e-5 for the other two models. We train each of these models for 3 epochs for both the datasets.

### 3.2.2 METRICS

The following metrics are calculated for both in-distribution ($I$) and out-of-distribution ($O$) evaluation sets:

- **S** : According to our null hypothesis, an unbiased model should predict the same label for both the $f$ and $m$ since they hold the same structure and differ only by a word (male/female). S is the % of instances where model gave the same prediction for both the cases. A low value for this metric is an indicator of high bias.

- $\Delta P$ : This represents the mean absolute difference between the entailment probabilities for the two hypothesis. According to our null hypothesis, a higher value is the indicator of high bias. This is the most important indicator of bias as it gives us the measure of difference in the prediction probabilities and helps us quantify the bias.

- **B** : A biased model would have higher entailment probability towards the hypothesis which represents the gender that an occupation is dominated by. For e.g., for the premise, *"The nurse is sitting on the bench"* where the profession is a female dominant one, a biased model will predict entailment with a higher probability when paired with a female-specific hypothesis. B is the % of instances where this bias is shown by the model. A higher value of B would indicate a higher bias but this should be observed along with $\Delta P$ to get a better understanding. A model with a higher value of B but lower $\Delta P$ would still be considered less biased when compared with a model with comparatively lower B but with a large mean absolute difference $\Delta P$.

### 3.3 RESULTS AND ANALYSIS

| | **SNLI (I)** | | | | **MNLI (I)** | | | |
|---|---|---|---|---|---|---|---|---|
| | Acc ($\uparrow$) | S ($\uparrow$) | $\Delta P(\downarrow)$ | B ($\downarrow$) | Acc ($\uparrow$) | S ($\uparrow$) | $\Delta P(\downarrow)$ | B ($\downarrow$) |
| BERT | 90.48 | 50.97 | 43.02 | **70.05** | 83.68 | 71.89 | 24.09 | 69.79 |
| RoBERTa | **91.41** | **72.13** | **27.23** | 77.79 | **87.59** | 64.35 | 21.85 | **65.12** |
| BART | 91.28 | 61.17 | 34.9 | 74.0 | 85.57 | **85.58** | **16.29** | 66.82 |

| | **SNLI (O)** | | | | **MNLI (O)** | | | |
|---|---|---|---|---|---|---|---|---|
| | Acc ($\uparrow$) | S ($\uparrow$) | $\Delta P(\downarrow)$ | B ($\downarrow$) | Acc ($\uparrow$) | S ($\uparrow$) | $\Delta P(\downarrow)$ | B ($\downarrow$) |
| BERT | 90.48 | 52.92 | 41.57 | **66.94** | 83.68 | 64.76 | 30.97 | 68.51 |
| RoBERTa | **91.41** | **71.02** | **26.57** | 73.76 | **87.59** | 64.33 | 24.86 | **64.53** |
| BART | 91.28 | 62.28 | 33.92 | 70.28 | 85.57 | **79.71** | **20.92** | 64.69 |

Table 5: Performance of the models when fine-tuned on SNLI and MNLI datasets respectively. The metric Acc indicates the model accuracy when trained on original NLI dataset (SNLI/MNLI) and evaluated on dev set (dev-matched for MNLI), S is the number of instances with same prediction: entailment or contradiction, $\Delta P$ denotes the mean absolute difference in entailment probabilities of male and female hypothesis and B denotes the number of times the entailment probability of the hypothesis aligning with the stereotype was higher than its counterpart (higher values for the latter two metrics indicate stronger biases). Numerics in bold represent the best value (least bias) for each metric. SNLI (O) and MNLI (O) represent the performance of various models fine-tuned on SNLI and MNLI respectively but tested on out-of-distribution evaluation set. SNLI (I) and MNLI (I) on the other hand have been tested on in-distribution evaluation set.

The main results of our experiments are shown in Table 5. For each tested model, we compute three metrics with respect to their ability to predict the correct label (Section 3.2.2). Our analysis indicate that all the NLI models tested by us are indeed gender biased.

From Table 5, metric B shows that all the tested models perform better when presented with pro-stereotypical hypothesis. However when observed along with $\Delta P$, we can see that BERT shows the most significant difference in prediction as compared to other models. Among the three models,

| | SNLI (I) | | | | MNLI (I) | | | |
| | $\Delta P$ ($\downarrow$) | | B ($\downarrow$) | | $\Delta P$ ($\downarrow$) | | B ($\downarrow$) | |
| | Male | Female | Male | Female | Male | Female | Male | Female |
|---|---|---|---|---|---|---|---|---|
| BERT | 51.43 | **33.6** | 98.19 | **37.22** | 27.16 | **20.51** | 95.23 | **40.11** |
| RoBERTa | 28.75 | **25.44** | 83.33 | **71.33** | 27.4 | **15.38** | 94.85 | **30.44** |
| BART | 35.22 | **34.54** | 89.14 | **56.33** | 16.99 | **15.46** | 90.47 | **39.22** |

| | SNLI (O) | | | | MNLI (O) | | | |
| | $\Delta P$ ($\downarrow$) | | B ($\downarrow$) | | $\Delta P$ ($\downarrow$) | | B ($\downarrow$) | |
| | Male | Female | Male | Female | Male | Female | Male | Female |
|---|---|---|---|---|---|---|---|---|
| BERT | 49.16 | **32.72** | 96.19 | **32.83** | 34.36 | **27.02** | 92.52 | **40.5** |
| RoBERTa | 28.46 | **24.35** | 78.76 | **67.94** | 30.58 | **18.18** | 93.8 | **30.38** |
| BART | 34.54 | **33.19** | 86.04 | **51.88** | 22 | **19.65** | 87.9 | **37.61** |

Table 6: Detailed analysis of how bias varies with respect to male and female dominated occupations. Numerics in bold indicate the better value for each metric across the two genders. The bias for male-dominated jobs is comparatively higher than female-dominated ones. Notations are same as those in Table 5.

BERT also has the lowest value of Metric S in almost all the cases, indicating highest number of label shifts thus showing the greatest amount of bias.

Figure 1: Comparison of trends in occupation bias across various models trained on original MNLI dataset. We compare the distribution of occupational-bias predicted by our models on in-distribution evaluation dataset (MNLI (I)) with the actual gender-domination statistics from CPS 2019. Models trained on SNLI also showed similar trends (Appendix A.2).

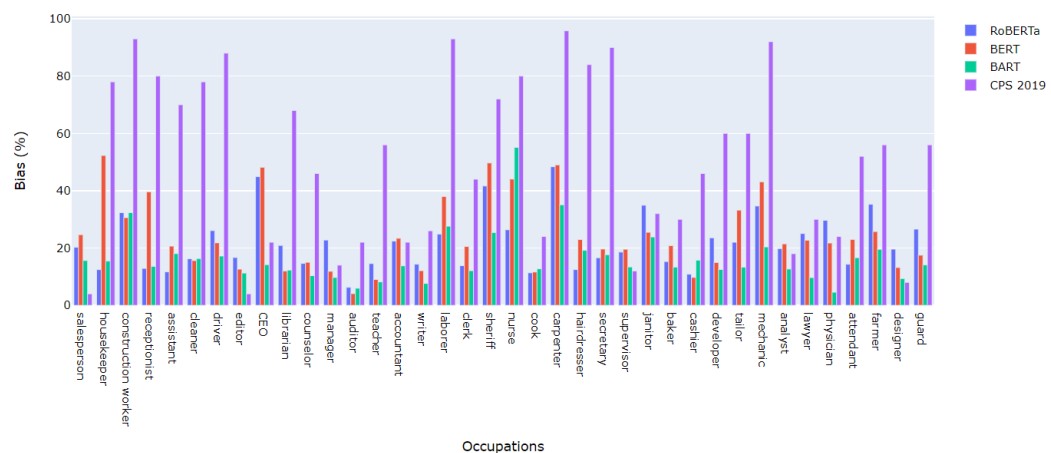

A detailed analysis with respect to gender can be found in Table 6, where we compare the values of B and $\Delta P$ when male dominated jobs are compared with female-dominated ones. From the results, it can be seen that the number of examples where the model was biased towards the pro-stereotypical hypothesis is higher for male-dominated occupations.

While both MNLI and SNLI show similar trends with respect to most metrics, results from Table 5 indicate that models fine-tuned on SNLI has a relatively higher bias than those trained on MNLI.

We also compare bias distribution across various occupations for the three models where the bias for the models (BERT, ROBERTa and BART) is the absolute difference between the entailment probabilities of two hypothesis. We compare this with CPS 2019 data where the difference between the gender distribution is used as the bias. Fig.1 shows that all the three models follow similar trends for occupational bias. We also compare this with the statistics on gender-domination in jobs given by CPS 2019 and validate that the bias distribution from models' predictions conforms with the real world distribution. Model predictions on occupations like nurse, housekeeper that are female dominated reflect higher bias as compared to those with equal divide, e.g. designer, attendant, accountant.

## 4 DEBIASING : GENDER SWAPPING

We follow a simple rule based approach for swapping. First we identify all the occupation based entities from the original training set. Next, we build a dictionary of gendered terms and their opposites (e.g. "his"↔"her", "he"↔"she" etc.) and use them to swap the sentences. These gender swapped sentences are then augmented to the original training set and the model is trained on it. Unlike Zhao et al's turking approach, we follow an automatic approach to perform this swapping thus our method can be easliy extended to other datasets as well. From Table 7, it can be seen the augmentation of training set doesn't deteriorate model performance (accuracy (Acc)) on the original training data (SNLI/MNLI). This simple method removes correlation between gender and classification decision and has proven to be effective for correcting gender biases in other natural language processing tasks. (Zhao et al. (2018a)).

| | SNLI (I) | | | | MNLI (I) | | | |
|---|---|---|---|---|---|---|---|---|
| | Acc ($\uparrow$) | S ($\uparrow$) | $\Delta P(\downarrow)$ | B ($\downarrow$) | Acc ($\uparrow$) | S ($\uparrow$) | $\Delta P(\downarrow)$ | B ($\downarrow$) |
| BERT | 90.50 | 57.17 | 35.02 | **67.02** | 84.04 | **87.48** | **14.60** | 72.07 |
| RoBERTa | **91.51** | 51.53 | 34.02 | 67.79 | **87.1** | 65.58 | 20.98 | **67.69** |
| BART | 90.6 | **61.89** | **31.71** | 72.76 | 85.76 | 75.33 | 21.32 | 70.67 |
| | SNLI (O) | | | | MNLI (O) | | | |
| | Acc ($\uparrow$) | S ($\uparrow$) | $\Delta P(\downarrow)$ | B ($\downarrow$) | Acc ($\uparrow$) | S ($\uparrow$) | $\Delta P(\downarrow)$ | B ($\downarrow$) |
| BERT | 90.50 | 65.5 | 30.01 | 66.71 | 84.04 | **78.76** | **19.99** | 72.53 |
| RoBERTa | **91.51** | 61.64 | 33.08 | **63.61** | **87.1** | 65.35 | 22.53 | **67.23** |
| BART | 90.6 | **66.43** | **27.56** | 70.05 | 85.76 | 68.43 | 26.51 | 68.67 |

Table 7: Performance of the models after debiasing. Notations are same as those in Table 5.

**Effectiveness of debiasing**: From results in Table 7, we can see that performance on BERT with respect to bias has improved following the debiasing approach. A comparison of the change in metrics $\Delta P$ and B before and after debiasing can also be seen in Fig. 2. (The figure here represents the trends with respect to in-distribution evaluation sets. Similar results were obtained for out-of-distrubution test and can be found in the Appendix (A.1))

The improvement in results for BERT indicate that maintaining gender balance in the training dataset is, hence, of utmost importance to avoid gender-bias induced incorrect predictions. The other two models, RoBERTa and BART, also show a slight improvement in performance with respect to most metrics. However, this is concerning since the source of bias in these cases is not the NLI dataset but the data that these models were pre-trained on. Through our results, we suggest that attention be paid while curating such dataset so as to avoid biased predictions in the downstream tasks.

## 5 RELATED WORK

**Gender Bias in NLP:** Rudinger et al. (2018) Prior works have revealed gender bias in various NLP tasks (Zhao et al. (2018a), Zhao et al. (2018c), Rudinger et al. (2017), Rudinger et al. (2018),

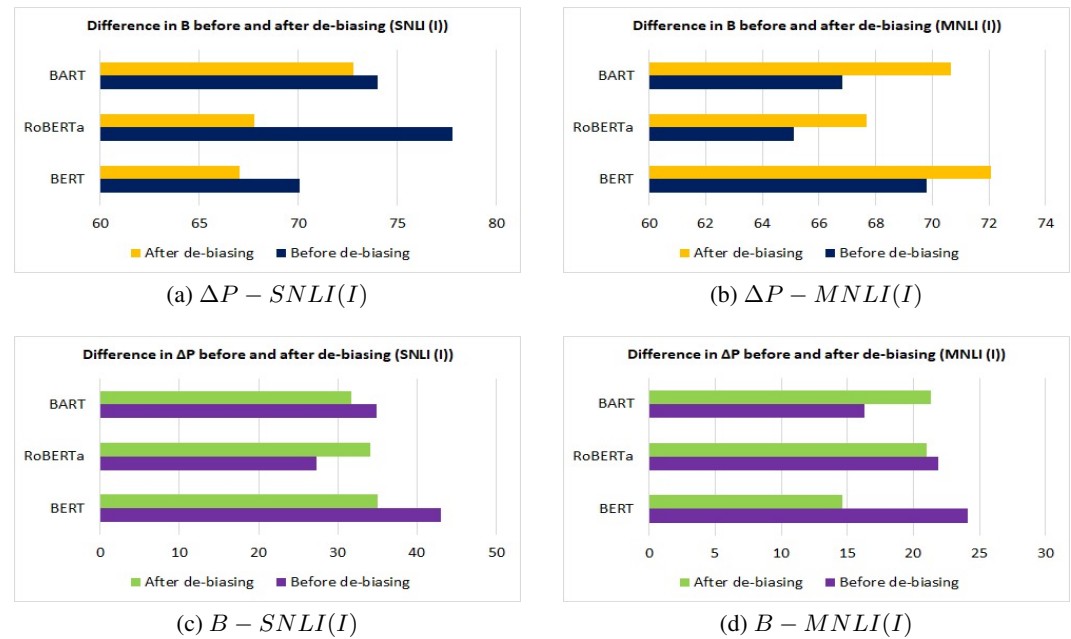

Figure 2: Difference in $\Delta P$ and $B$ in MNLI and SNLI in-distribution evaluation sets before and after de-biasing. Similar results are observed for out-of-distribution evaluation sets (Appendix A.1).

Caliskan et al. (2017), Gaut et al. (2020), He et al. (2019a), Bolukbasi et al. (2016)). Authors have created various challenge sets to evaluate gender bias, for example Gaut et al. (2020) created WikiGenderBias, for the purpose of analyzing gender bias in relation extraction systems. Caliskan et al. (2017) contribute methods for evaluating bias in text, the Word Embedding Association Test (WEAT) and the Word Embedding Factual Association Test (WEFAT). Zhao et al. (2018a) introduced a benchmark WinoBias for conference resolution focused on gender bias. To our knowledge, there are no evaluation sets to measure gender bias in NLI tasks. In this work, we fill this gap by proposing a methodology for the same.

**Data Augmentation:** Data augmentation has been proven to be an effective way to tackle challenge sets (Andreas (2020a), McCoy et al. (2019), Jia & Liang (2017), Gardner et al. (2020)). By correcting the data distribution, various works have shown to mitigate bias by a significant amount Min et al. (2020), Zhao et al. (2018a), Glockner et al. (2018). In this paper, we use a rule-based gender swap augmentation similar to that used by Zhao et al. (2018a).

## 6    CONCLUSION AND FUTURE WORK

We show the effectiveness of our challenge setup and find that a simple change of gender in the hypothesis can lead to a different prediction when tested against the same premise. This difference is a result of biases propagated in the models with respect to occupational stereotypes. We also find that the distribution of bias in models' predictions conforms to the gender-distribution in jobs in the real world hence adhering to the social stereotypes. Augmenting the training-dataset in order to ensure a gender-balanced distribution proves to be effective in reducing bias for BERT indicating the importance of maintaining such balance during data curation. The debiasing approach also reduces the bias in the other two models (RoBERTa and BART) but only by a small amount indicating that attention also needs to be paid on the dataset used for training these language models. Through this work, we aim to establish a baseline approach for evaluating gender bias in NLI systems, but we hope that this work encounters further research by exploring advanced debiasing techniques and also exploring bias in other dimensions.

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
