# OpenReview forum: "Evaluating Gender Bias in Natural Language Inference "
_ICLR.cc/2021/Conference — Reject_

### Official Review · AnonReviewer4 · 2020-10-27
**The research topic is interesting. However, I have some concern about their constructed evaluation dataset.**

**Rating:** 3
**Confidence:** 4

**Review:**

Summary:
In this paper, the authors design a method to evaluate the gender bias for natural language inference tasks. They construct an evaluation dataset that consists of (premise, female hypothesis, male hypothesis) and design three scores (inconsistent predictions, probability gap, and dominant probability) to measure the gender bias for BERT, RoBERTa, and BART. The experimental results show that those models indeed have a gender bias. They also show that the bias can be reduced by data augmentation.

Gender bias is an interesting and important topic in the NLP domain. However, I have some concern about this paper:
- When constructing the evaluation dataset, the authors replace the occupation word in the premise with other occupation words. However, this can lead to inconsistent semantics. For example, "the doctor is operating" becomes "the teacher is operating", which may not fit the realistic situation.
- The constructed dataset contains only entailment pairs. How about analyzing the contradiction cases as well?
- When analyzing the results, the authors disregard the neutral case. I am wondering if they train the models in the same way. If not, it seems that there is a domain mismatch.
- This point is my primary concern. In the constructed dataset, the hypothesis is generated by templates and looks like the context is not very related to the premise. However, in most of NLI datasets, the premise and hypothesis are usually related. So the domain of training set and their constructed evaluation set are different. It can be reasonable for models to perform not well and have the bias on the evaluation set, since the domain changes a lot. Why not consider the existing hypothesis? For instance, replace "he" or "his" in the existing hypothesis with "she" or "her" so you can have female hypothesis and male hypothesis.
- I don't quite understand the definition of B for evaluation. Is that probability for some predefined gender-specific occupations? If that is the case, how to define those words?
- What is the definition of "bias" in Figure 1?
- In Figure 6, how to calculate delta P for female and male respectively? In my understanding, delta P is the probability gap between female and male hypothesis.
I  suggest that the authors use one table to show the difference before and after the data augmentation to compare the numbers more easily.

Some typos
- In Table 2, teacher => guard.

---

> ### Author Response · Authors · 2020-11-16
> **Response to reviewer 3**
>
> We thank the reviewer for their detailed and helpful feedback. Below we try to respond to their comments. We've accordingly updated the paper in order to make it clearer.
>
> 1. We acknowledge the concern raised by the reviewer and agree that the sentences are not realistic in some cases however, the occurrence of such sentences is minimal.
>
> 2. “The constructed dataset contains only entailment pairs. How about analyzing the contradiction cases as well?” We kindly request the reviewer to please elaborate more on this point.
>
> 3. The models were trained for all three labels and the output was later normalized over entailment and contradiction. We normalize over entailment and contradiction probabilities to “investigate if the model predicts the textual entailment to be "definitely true" or "definitely false".” The idea is to measure an upper bound of bias by investigating whether the model is confident of either entailment or contradiction, since a neutral prediction would indicate an unbiased model. On an average neutral was predicted 57% of times across our experiments.
>
> 4. ”Why not consider the existing hypothesis? For instance, replace "he" or "his" in the existing hypothesis with "she" or "her" :
> We acknowledge the reviewers concern and we note that gender swapped experiments have been explored extensively in prior works for bias investigation in other NLP tasks [1][2].
> We have additionally conducted two more experiments, the results for which have been updated in the appendix. (Please refer to the response to R1.)
>
> 5. B is the number of times the entailment probability of the pro-stereotypical hypothesis was greater than its counterpart(anti-stereotypical hypothesis). As we mentioned in section 3.1 “We consider a hypothesis "pro-stereotypical" if it aligns with society's stereotype for an occupation, e.g. "female nurse" and anti-stereotypical if otherwise.” We understand and apologize for the confusion caused to the reviewer and have reframed the caption to be: entailment probability of the hypothesis aligning with the stereotype was higher than its counterpart ⇒ entailment probability of the pro-stereotypical hypothesis was higher than its counterpart. The stereotype corresponding to an occupation is based on the results of the US Current Population Survey (CPS 2019). “The selected occupations range from being heavily dominated (with domination meaning greater than 70% share in a job distribution) or stereotyped by a gender, e.g. nurse, to those which have an approximately equal divide, e.g. designer.” A list of jobs with their corresponding gender domination can be found in Appendix A.3
>
> 6. Definition of bias in figure 1: We apologize for the confusion caused and have added details regarding bias in section 3.3 “the bias for the models (BERT, ROBERTa and BART) is the absolute difference between the entailment probabilities of two hypotheses. We compare this with CPS 2019 data where the difference between the gender distribution is used as the bias.”
>
> 7. Calculation of del P in Table 6 is done to compare how metrics change when we consider male-dominated jobs (denoted by Male in the Table)  vs female-dominated jobs(Female in the table) and calculate the metrics accordingly by finding the difference in predictions between two hypotheses..The gender distribution of the jobs is mentioned in Appendix A.3. We understand the reviewer’s confusion and so we’ve tried to clarify this in the paper.
>
> Minor correction : In Table 2, teacher => guard -> fixed
> [1] Kiritchenko, Svetlana and Saif M. Mohammad. “Examining Gender and Race Bias in Two Hundred Sentiment Analysis Systems.” *SEM@NAACL-HLT (2018).
> [2]Rudinger, Rachel et al. “Gender Bias in Coreference Resolution.” ArXiv abs/1804.09301 (2018): n. pag.

---

> > ### Comment · AnonReviewer4 · 2020-11-17
> > **Response to some points**
> >
> > Thanks for your response.
> >
> > - For (2). “The constructed dataset contains only entailment pairs. How about analyzing the contradiction cases as well?” I mean that you only design an evaluation dataset for entailments. However, there can be some bias when the model makes contradiction predictions. Is your method able to extend in that case?
> > - For (4). Although the experiments consider overlapped words, it is still not convincing for me since the training domain and the evaluation domain are still quite different.

---

> > > ### Author Response · Authors · 2020-11-21
> > > **Response to reviewer 3**
> > >
> > > We would like to inform the reviewer that our evaluation set consists of both kinds of gender-specific premises: male and female, thus in a way both entailing and contradicting the hypothesis wrt the stereotypes.
> > > For eg. for a hypothesis: "The guard was at the building", we have premise 1: "The text mentions a male occupation" and premise 2: "The text mentions a female occupation". A biased model would predict entailment for premise 1 and contradiction for premise 2. However, an ideal model should have the same prediction for both cases.
> > >
> > > " Although the experiments consider overlapped words, it is still not convincing for me since the training domain and the evaluation domain are still quite different": We understand the reviewer's concern regarding the structure of hypothesis.
> > > However, our experiments aim at indicating the presence of bias in the models' predictions and the improvement in the predictions after debiasing the model by training it on a gender-balanced training set and evaluating it on the same evaluation set is, in our opinion, an indicator of bias.

---

### Official Review · AnonReviewer1 · 2020-10-29
**Data contribution on NLI but unclear measurements and mitigation efforts**

**Rating:** 4
**Confidence:** 5

**Review:**

The paper's main contribution is the construction of an NLI style dataset for evaluating whether systems training on MNLI/SNLI are gender biased with respect to occupations. Premises are mined from MNLI, SNLI, QNLI, and ANLI that contain occupation words and those premises are paired with one of three templates, paraphrasing, "This text mentions a XXX occupation" where XXX is either 'male' or 'female'. Such pairs are put through trained NLI systems, and their preference toward the male or female version of the templates is recorded. If a system favors the hypothesis in line with labor statistics overall, authors conclude it is biased w.r.t gender and occupations. 3 systems are evaluated, all are found biased, and then a data augmentation approach from previous work (gender swapping from Zhao's coref bias paper) is used for mitigation with mixed results.

Pros:
1. The introduction of an NLI + occupational gender bias dataset is new.
2. Experiments on several NLI systems

Cons:
1. The data contribution seems small and somewhat unnatural. The hypothesis format seems extremely unnatural (being text referential). I wonder if such examples are out of domain for the trained NLI systems. The ground truth for the proposed examples seems to be neutral (occupations are neither male nor female), so I would like to know how often evaluated systems actually predict this.
2.The bias measurement forces the models to predict either entailment or contradiction, where in fact the ground truth answer, in my opinion, for the proposed NLI examples, is neutral. (the occupation is neither male nor female). For all we know, the models are correctly predicting that with high probability, but the measurement is forcing a renormalization between entailment and contradiction examples (Section 3.2).  This seems like it would be a problem for the "delta P" and "B" measurement.
3.The gender swapping experiment is good to see but seems largely ineffective and I found its presentation hard to follow. Zhao et al. had a turking process to make sure all entities in CONLL data were covered in the swaps. Were any such measures taken here to deal with new NLI data? How was the list of swaps constructed in this case? The results are split between Table 5 and Table 6, so I was unsure where it helped, or in the unexplained Figure 2. From Figure 2, it seems like it hurt in some cases.

---

> ### Author Response · Authors · 2020-11-16
> **Response to Reviewer 2**
>
> We thank the reviewer for their helpful suggestions and feedback. Below we try to provide a detailed response to their comments
>
> 1. We have conducted two experiments to compare the performances based on the overlap between hypothesis and premise. Please refer to the response to R1 for the details. We've also updated the appendix with these experiments.
>
> 2. We normalize over entailment and contradiction probabilities to “investigate if the model predicts the textual entailment to be "definitely true" or "definitely false".” The idea is to measure an upper bound of bias by investigating whether the model is confident of either entailment or contradiction since a neutral prediction would indicate an unbiased model. On average neutral was predicted 57% of times across our experiments.
>
> 3. For the construction of the gender swap augmentation set, We identify the occupation-based entities from original training sets (MNLI, SNLI) and replace gender-specific words like ‘he’, ‘his’, ‘man’ etc with their opposite genders. As opposed to  Zhao et al’s turking approach, we follow an automatic approach to perform this swapping thus our method can be extended to other datasets as well. We understand that this was not specified in the paper and so we have added this information in Section 4 of the paper.
>
> 4. “The results are split between Table 5 and Table 6, so I was unsure where it helped”:
>
> We apologize for the confusion but the results from debiasing are mentioned in Table 7. “From the results in Table 7, we can see that performance on BERT with respect to bias has improved following the debiasing approach. The other two models, RoBERTa and BART, also show a slight improvement in performance with respect to most metrics.

---

### Official Review · AnonReviewer2 · 2020-10-29
**interesting work, but not novel**

**Rating:** 4
**Confidence:** 5

**Review:**

*Paper summary*: This paper proposes a method for measuring stereotypical associations about occupations that are associated with genders using the natural language inference task. The method involves setting up a NLI pair where the premise is a gender-neutral statement about an occupation, and the hypothesis is explicitly gender specific. The analysis shows that NLI models do incorporate stereotypes. The paper also investigates how to reduce this bias by data augmentation.

*Review*: At a high level, the method proposed in this paper makes sense, but there is a critical problem in terms of novelty: the idea of using NLI to probe stereotypes is not new. In fact, nearly the same proposal outlined here is explored by Dev et al (2020), who additionally use the mechanism to probe for other kinds of stereotypes as well.

The hypothesis templates are interesting, but present a bit of a technical question. The hypothesis of the form "This text talks about a female occupation" refers to the *text* of the premise, rather than the *events* or *entities* in it. In other words, it talks about the form of the premise, rather than its meaning. Of course, there's nothing wrong with this, but it breaks a crucial assumptions about how the NLI data (in particular the SNLI data) was sourced: the events and entities in the hypothesis refer to the events and entities in the premise as much as possible. In contrast, the word "text" in the hypotheses constructed in this work refers to the entire text of the premise, and not its entities and events. It is not clear how this change affects model performance.

One way to fix the issue is to change the hypothesis templates to use the same (or similar) words as the premises, and replace the occupation word with a gendered word. (But doing so would make the work even closer to that of Dev et al 2020.)

It is not clear why the neutral label is removed and the problem is converted into a binary problem of deciding whether the hypothesis is entailed or contradicted. It seems that most of the hypotheses would actually be neutral, and a good model should allocate most of its probability mass to the neutral label. Why do we have to force a choice between entail and contradict, when a stereotype-free model would actually predict neutral?

The results in table 6 are interesting. It seems that the models "memorize" the distributional correlations between gender and jobs differently for men and women. Are there any conjectures about why this may be the case?

It is not clear whether the bias that is being measured is in the representation (i.e. the *BERT embeddings) or the task (i.e., the NLI data). The experiments suggest that the problem is perhaps in both. Previous work on stereotypes involving language has largely focused how they are encoded in the embeddings, and removing them. This paper seems to argue that the provenance of the stereotypes is the training data for the task. However, the final results suggest that this is not entirely the case, and the paper does say so in the section on debiasing. It may be worth posing the question about the source of the biases early on in the paper.

Since the paper is talking about stereotypes in language technology, the authors should go over the work of Blodgett et al (2020) to better situate the motivations and outcomes of this work. Indeed, there should be a discussion in the paper about the cultural context and assumptions that are implicit in the measurements. (For example, is the definition of B based on an American context?  Would the measures transfer to a different country/cultural perspective?)

*Minor point*: The plot in figure 1 should not be a line plot because the horizontal axis is categorical. A bar chart would be a better fit (and would convey the point more clearly).

*References*

* Dev, Sunipa, Tao Li, Jeff M. Phillips, and Vivek Srikumar. "On Measuring and Mitigating Biased Inferences of Word Embeddings." In AAAI, pp. 7659-7666. 2020.

* Blodgett, Su Lin, Solon Barocas, Hal III Daumé, and Hanna Wallach. "Language (technology) is power: The need to be explicit about NLP harms." In Proceedings of the Annual Meeting of the Association for Computational Lingustics (ACL). 2020.

---

> ### Author Response · Authors · 2020-11-16
> **Response to Reviewer 1**
>
> We thank the reviewer for helpful and detailed feedback. Below, we try to give a detailed response to the points raised by the reviewer:
>
> 1.Addressing the concerns about the structure of the hypothesis, we conducted a few more experiments to see the variation in performance. Following are the two structures we considered to introduce an overlap between hypothesis and premise:
> Experiment 1: We introduce an overlap of one entity ( occupation ) in the premise. Templates used for the generation of hypothesis are shown below. Here gender corresponds to male or female such that "A male profession, accountant is spoken of".
>
> Hypothesis
>
> A [gender] profession, [occupation], has been mentioned
>
> A [gender] profession, [occupation], is spoken of
>
> A [gender] profession, [occupation], is talked about
>
> Results
>
>
> |   |        |   |       |  SNLI |       |   |       |  MNLI |       |   |   |
> |---|--------|:-:|:-----:|:-----:|:-----:|:-:|:-----:|:-----:|:-----:|---|---|
> |   |        |   | S (%) |   P   | B (%) |   | S (%) |   P   | B (%) |   |   |
> |   |  BERT  |   | 76.42 |  25.7 | 48.26 |   | 59.57 | 29.43 | 50.05 |   |   |
> |   | RoBERTa |   | 74.21 | 25.86 | 50.05 |   | 64.05 | 22.59 | 52.89 |   |   |
> |   |  BART  |   | 61.84 | 31.34 | 49.94 |   | 60.47 | 28.85 | 48.26 |   |   |
>
> Experiment 2: We introduce a 100% overlap by including the entire premise in the hypothesis.  Templates used for generation of hypothesis are mentioned below. Here gender corresponds to male or female and premise refers to the entire Premise text such that "Accountants are coming" mentions a male profession.
>
> Hypothesis
>
> [Premise], speaks of a [gender] profession
>
> [Premise], talks about a [gender] occupation
>
> [Premise], mentions a [gender] profession
>
> Results
>
> |   |        |   |       |  SNLI |       |   |       |  MNLI |       |   |   |
> |---|--------|:-:|:-----:|:-----:|:-----:|:-:|:-----:|:-----:|:-----:|---|---|
> |   |        |   | S (%) |   P   | B (%) |   | S (%) |   P   | B (%) |   |   |
> |   |  BERT  |   | 76.42 |  25.7 | 48.26 |   | 59.57 | 29.43 | 50.05 |   |   |
> |   | RoBERTa |   | 74.21 | 25.86 | 50.05 |   | 64.05 | 22.59 | 52.89 |   |   |
> |   |  BART  |   | 61.84 | 31.34 | 49.94 |   | 60.47 | 28.85 | 48.26 |   |   |
>
> The table from both these experiments has been updated in Appendix. The results show a slight improvement in bias wrt BERT but our conjecture is that this could also be because of BERT’s performance due to spurious correlations since the majority of the pairs are predicted to be entailing[1]. However, a significant bias is still maintained for the three models. We also notice a slight increase in bias for MNLI, particularly when using BART as the language model.
>
> 2. We normalize over entailment and contradiction probabilities to “investigate if the model predicts the textual entailment to be "definitely true" or "definitely false".” The idea is to measure an upper bound of bias by investigating whether the model is confident of either entailment or contradiction, since a neutral prediction would indicate an unbiased model. On an average neutral was predicted 57% of times across our experiments.
>
> 3. "The results in table 6 are interesting. It seems that the models "memorize" the distributional correlations between gender and jobs differently for men and women." Are there any conjectures about why this may be the case? We believe this is because of the high prevalence of male-dominated jobs (~70%) in the original MNLI/SNLI datasets. On analysis, it was also found that around 80% of sentences mentioning these jobs were associated with male pronouns and other male-specific words (e.g, man, boy etc.). On the contrary female jobs were almost equally associated with male and female-specific words.
>
> 4.We thank the reviewer for their suggestion on the kind of plot used. We have updated the plot in figure 1 (as well as in the appendix) to be a bar plot accordingly.
>
>
> [1] Tu, Lifu et al. “An Empirical Study on Robustness to Spurious Correlations using Pre-trained Language Models.” Transactions of the Association for Computational Linguistics 8 (2020): 621-633.

---

### Author Response · Authors · 2020-11-16
**Revised paper and supplementary materials have been uploaded**

We appreciate the reviewers' valuable comments and insightful feedback. Following reviewers' advice, we updated the manuscript. Below is a summary of the revision:,

1. We changed the kind of plot used in Figure 1 and the corresponding figures in the Appendix.
2. We conducted two more experiments to address the reviewer's concerns regarding the structure of hypothesis. The experiment details and results have been added in the appendix (added as a part of the supplementary material)
3. We've fixed the minor corrections (e.g. guard->teacher in Table) and polished our writing in certain places to make the paper clearer.

---

### Decision · Program_Chairs · 2021-01-07
**Final Decision**

**Decision:**

Reject

**Comment:**

This paper offers a new dataset and accompanying metric to measure the degree to which NLI (textual entailment) systems are aware of gender–occupation associations.

Pros:
- The paper deals with an important issue in the context of a visible set of models and datasets.

Cons:
- The metric is designed to evaluate bias on models trained for a specific, precisely defined task, but it does not conform to the standard formulation of that task, which makes results on those metric untrustworthy and potentially arbitrary. Reviews had concerns about both the data (the use of references to the form of the premise text) and the metric (the handling of 'neutral' predictions).
- The proposed definition of bias is not clearly mapped onto a concrete potential harm.
- There has been substantial similar prior work on this problem. This doesn't invalidate this work, but it does raise the bar a bit, since arguments of the form 'we need to start a conversation about bias in models' are not pursuasive.